# Genomic and In Vitro Phenotypic Comparisons of Epidemic and Non-Epidemic Getah Virus Strains

**DOI:** 10.3390/v14050942

**Published:** 2022-04-30

**Authors:** Noor-Adila Mohamed-Romai-Noor, Sing-Sin Sam, Boon-Teong Teoh, Zur-Raiha Hamim, Sazaly AbuBakar

**Affiliations:** 1Tropical Infectious Diseases Research and Education Centre, Universiti Malaya, Kuala Lumpur 50603, Malaysia; nooradila@um.edu.my (N.-A.M.-R.-N.); boonteong@um.edu.my (B.-T.T.); zur.raiha@um.edu.my (Z.-R.H.); 2Institute for Advanced Studies, Universiti Malaya, Kuala Lumpur 50603, Malaysia

**Keywords:** alphavirus, Getah virus, infectious diseases, emerging, tropical, arbovirus

## Abstract

Getah virus is an emerging mosquito-borne animal pathogen. Four phylogenetic groups of GETV, Group I (GI), GII, GIII and GIV, were identified. However, only the GETV GIII was associated with disease epidemics suggesting possible virulence difference in this virus group. Here, we compared the genetic and in vitro phenotypic characteristics between the epidemic and non-epidemic GETV. Our complete coding genome sequence analyses revealed several amino acid substitutions unique to the GETV GIII and GIV groups, which were found mainly in the hypervariable domain of nsP3 and E2 proteins. Replication kinetics of the epidemic (GIII MI-110 and GIII 14-I-605) and non-epidemic GETV strains (prototype GI MM2021 and GIV B254) were compared in mammalian Vero cells and mosquito C6/36 and U4.4 cells. In all cells used, both epidemic GETV GIII MI-110 and GIII 14-I-605 strains showed replication rates and mean maximum titers at least 2.7-fold and 2.3-fold higher than those of GIV B254, respectively (Bonferroni posttest, *p* < 0.01). In Vero cells, the epidemic GETV strains caused more pronounced cytopathic effects in comparison to the GIV B254. Our findings suggest that higher virus replication competency that produces higher virus titers during infection may be the main determinant of virulence and epidemic potential of GETV.

## 1. Introduction

Getah virus (GETV) is a mosquito-borne virus that belongs to the genus *Alphavirus* in the family of *Togaviridae* [1]. It is enveloped and spherical with a diameter of approximately 70 nm [2] and contains a single-stranded positive-sense RNA genome of 11–12 kb in length. The genome consists of two open reading frames that encode four non-structural proteins (nsP1, nsP2, nsP3 and nsP4), which are responsible for viral RNA transcription and replication, and five structural proteins (capsid protein C, glycoproteins E3, E2, E1, and 6K), which are responsible for viral binding and entry into host cells during infection [1,3].

The first GETV strain, MM2021, was isolated in Malaysia in 1955 from *Culex gelidus* [4]. Currently, GETV is present throughout the East and Southeast Asia, as well as in Northern Australia [5,6,7]. Mosquitoes of *Culex* and *Aedes* species are the main vectors for the transmission of GETV [6,8]. Serological evidence of GETV infection has been reported in a wide range of vertebrate hosts including birds, reptiles, and mammals, and humans [9].

GETV has become one of the emerging animal pathogens that poses increased health threat to racehorses and pigs. Several outbreaks of epizootic diseases have been reported in these animals in Japan, China and India causing great economic losses [2,10,11,12,13,14,15]. The disease in horses is generally self-limiting, present with fever, anorexia, hind limb edema and stiff gaits [13]. The GETV infection, on the other hand, caused severe and fatal diseases in young piglets, and reproductive failure in pregnant sows that lead to stillbirths and fetal deaths [15]. Recently, GETV infection has also been associated with neurological symptoms and death in blue foxes and fever in cattle [16,17]. GETV infection, however, is not known to cause any disease in humans.

Phylogenetic analyses of all known GETVs have identified four major lineages of viruses, designated as Group I (GI), GII, GIII, and GIV [18]. The GI and GII consist of the old GETV isolates, the Malaysia GETV MM2021 (1955) and Japan Sagiyama virus (1956), respectively, while the GIII and GIV comprise the most recent circulating virus strains. Currently, the GIII lineage is the dominant lineage with the largest virus populations, comprising mainly the virus strains associated with the animal disease epidemics. The GIV lineage, however, is comprised mostly of viruses that were found in the mosquitoes, including the recent Malaysian GETV B254 (2012) discovered in our previous study [19]. Recently, one GIV strain (GETV/SW/Thailand/2017) was isolated from pig serum in Thailand in 2017, with no clinical signs reported [20]. Nevertheless, it remains uncertain whether the GIV viruses will be competent for epidemic spreading in the future. 

Currently, the GIII remains the only lineage that is associated with pathogenesis and animal disease epidemics. This is possibly attributed to the different viral fitness or virulence characteristics given by the specific variations in the genetic makeup of the viruses. Thus, in this study, we examined and compared the complete coding genome sequences and in vitro replication competence of the epidemic and non-epidemic GETV groups in mammalian and mosquito cell lines. Here, we report several amino acid substitutions specific to the GETV GIII and GIV viruses in the nsP3 and E2 genes, which may play a role in the higher replication competency of the epidemic GIII viruses, compared to the non-epidemic GI and GIV GETV.

## 2. Materials and Methods

### 2.1. Cell Culture

Two *Aedes albopictus* mosquito cell lines, C6/36 and U4.4 (ATCC), and one mammalian cell line, Vero (ECACC), were used in this study. While the mammalian Vero cells and the *Aedes albopictus* C6/36 mosquito cells are common susceptible cell lines used for alphavirus propagation and replication studies, the U4.4 is an RNAi-competent mosquito cell line that could be a representative model for GETV infection in mosquitoes in nature. The C6/36 cells were cultured in Eagle’s Minimum Essential Medium (EMEM) (HyClone, Logan City, UT, USA) supplemented with 10% fetal bovine serum (FBS) (Thermo Fisher Scientific, Waltham, MA, USA), 2 mM of l-glutamine, and 0.1 mM of 1× non-essential amino acids (NEAA). The cells were incubated at 28 °C in 3% CO_2_. The U4.4 was cultured in Leibovitz’s L-15 media (Sigma-Aldrich, Burlington, MA, USA) supplemented with 10% FBS, 8% Tryptose Phosphate Broth (Sigma-Aldrich, Burlington, MA, USA), and 25 µg of streptomycin/penicillin. The U4.4 cells were incubated at 28 °C without CO_2_.

The Vero cells were cultured in Dulbecco’s Modified Eagle’s Medium (DMEM) (HyClone, Logan City, UT, USA) containing 10% FBS, 2 mM of l-glutamine, and 0.1 mM of 1× NEAA. The cells were incubated at 37 °C in 5% CO_2_.

Culture medium supplemented with 2% FBS, 2 mM of l-glutamine, 0.1 mM 1× NEAA, and 25 µg/mL of streptomycin/penicillin were used as the maintenance medium for respective cell lines during the GETV infections. 

### 2.2. Getah Viruses

Four GETV strains were used in this study; two mosquito-origin GETV strains from Malaysia (MM2021 and B254) and two equine-origin GETV strains from Japan (MI-110 and 14-I-605). Strain MM2021 was provided by the World Reference Center for Emerging Viruses and Arboviruses, The University of Texas Medical Branch, Galveston, TX, USA. Strain B254 was isolated from *Culex fuscocephala* in Peninsular Malaysia between 2011–2014 [19]. The epidemic strains, MI-110 and 14-I-605, were isolated from infected equines during the GETV outbreaks in Japan in 1978 and 2014, respectively [5]. Both epidemic strains were provided by Dr. Hiroshi Bannai and Dr. Manabu Nemoto from Equine Research Institute, Japan Racing Association, Tochigi, Japan. The MM2021 and B254 represent GI and GIV, respectively [18,19], while both MI-110 and 14-I-605 represent GIII GETVs [18].

### 2.3. Sequence Comparison of GETV Strains

Whole genome sequences of all GETVs available in GenBank were downloaded and aligned using Clustal X version 2.0 software. A sequence alignment based on the complete coding region was generated using GeneDoc version 2.7 software [21] and subjected to nucleotide and amino acid sequence analyses using the GeneDoc and BioEdit version 7.2.5 [22] software.

### 2.4. Infection of Cells with Different GETV Strains

All three cell lines, Vero, C6/36, and U4.4, were seeded in a 96-well plate at a concentration of 2 × 10^4^ cells/100 µL/well in the maintenance media. Cells were incubated overnight at appropriate culture temperature and CO_2_ conditions, as mentioned previously in Section 2.1, for cell attachment. Each cell line was then infected with different GETV strains at multiplicity of infection (MOI) of 0.1. Infection was performed in triplicates. Cells were incubated at room temperature for 1 h with gentle rocking before the inoculum was replaced with maintenance media. Infected cell culture supernatants were harvested at 0, 8, 24, 48, 72, and 96 hours post infection (hpi). One hundred and forty microliters of the cell culture supernatants were subjected to viral RNA extraction using QIAmp Viral RNA Mini Kit (Qiagen, Hilden, Germany) according to manufacturer’s protocol. The viral RNA was eluted in 60 µL of RNase-free water and kept at −80 °C until used.

### 2.5. Viral RNA Quantitation Using TaqMan^®^ Probe-Based qRT-PCR

The GETV RNA titer was quantitated using an in-house established TaqMan^®^ probe-based quantitative reverse transcription-polymerase chain reaction (qRT-PCR) assay as previously described [23]. The qRT-PCR was performed in a total of 10 µL in a reaction containing 5.0 µL of 2× SensiFAST Probe Hi-ROX One-Step Mix, 0.22 µL of probe/primer, 0.1 µL of Reverse Transcriptase, 0.2 µL of RiboSafe RNase Inhibitor, 1.33 µL of RNA, and 3.15 µL of DEPC-treated water. The qRT-PCR was performed using an Applied Biosystem StepOne Real-Time PCR System (Thermo Fisher Scientific,, Waltham, MA, USA) with a thermal profile as follows: 45 °C for 10 min; 95 °C for 2 min; and 40 cycles of 95 °C for 5 s and 60 °C for 20 s. Titers of GETV in the supernatants were determined based on a standard curve generated using serial dilutions of the GETV RNA standard that ranged from 10^7^–10^1^ RNA copies/µL. 

### 2.6. Plaque Assay

Vero cells were seeded in a 24-well plate at a concentration of 2 × 10^5^/500 µL/well in DMEM supplemented with 10% FBS, 2 mM of l-glutamine, and 0.1 mM of 1× NEAA, and incubated overnight. The medium was removed from each well and replaced with 200 µL of viral inoculum mixed with serum-free media in a 1/10 dilution. Plates were left to rock for 1 h at room temperature. The inoculum was discarded and replaced with 1 mL of carboxy methyl cellulose (CMC) in DMEM containing 2% FBS, 2 mM of l-glutamine, and 0.1 mM of 1× NEAA. The plates were incubated at 37 °C in 5% CO_2_ for 3 days before cell fixing and staining with 4% paraformaldehyde (PFA) and 1% crystal violet mixed in 20% EtOH, respectively.

### 2.7. Statistical Analyses

The replication growth curves of the GETV strains in respective cell lines were plotted and analyzed with two-way ANOVA and linear regression analyses. The replication rate of GETV was estimated by determining the slope of the linear regression curve. The Bonferroni posttest was performed to determine significant differences of the mean titers attained between GETV strains. All statistical analyses were performed using Graph Pad Prism 5 (Graph Pad Software Inc., San Diego, CA, USA).

## 3. Results

### 3.1. Sequence Analyses of Different GETV Groups 

Multiple sequence alignments comprised of the complete coding sequences of GETV strains were generated and a phylogenetic tree constructed for the different GETV groups (Appendix A). The nucleotide and amino acid sequences of the different GETV groups (GI, GII, GIIIa-GIIIe, GIV) were analyzed using the GeneDoc version 2.7 [21] and BioEdit version 7.2.5 [22] software. Comparisons of the amino acid sequences at the non-structural and structural proteins of GETV GI, GIII, and GIV viruses against the GETV Sagiyama revealed amino acid substitutions exclusive to the GIII and GIV viruses. Distinct amino acid substitutions in the nsP1, nsP2, nsP3, C, E2, and E1 genes were noted within the GETV GIII (Figure 1 and Table 1). Two non-conservative amino acid substitutions, T461P and G467E, were found in the hypervariable carboxyl-terminal (*C*-terminal) of nsP3 of the GIII viruses. On the other hand, amino acid substitutions specific to the GIV Malaysian GETV B254, China GETV YN12031, and GETV/SW/Thailand/2017 were observed in the nsP2, nsP3, C, E3 and E2 proteins. In contrast to the GIII viruses, these GIV strains accumulated more non-conservative amino acid substitutions (n = 7), which were H374Y (nsP2), D386A (nsP3), P466R (nsP3), W501Q (nsP3), T505I (nsP3), D109M (E2) and S205N (E2), when compared against the GII Sagiyama strain. The GIV Russia LEIV16275 Mag, on the other hand, showed rather different amino acid substitutions in comparison to the other GIV strains.

### 3.2. Plaque Morphology of Different GETV Strains

In this study, four GETV strains, GI MM2021, GIII MI-110, GIII 14-I-605, and GIV B254 were used and compared for their in vitro replications. The virus inoculums were prepared using C6/36 cells and virus titers were determined by plaque assays using Vero cells. The plaque sizes were measured across four independent experiments and *t*-test was used to compare means. All GETV strains used produced distinct plaques of heterogenous sizes (Figure 2). In general, the sizes of the plaques formed by GIV B254, ranging between 9.3–28.2 mm, were significantly smaller than those of GI MM2021 (24.8–50.1 mm) (*p* < 0.01), GIII MI-110 (11.6–49. 8 mm) (*p* < 0.05), and GIII 14-I-605 (28.7–63.1 mm) (*p* < 0.01). 

### 3.3. Replication Competencies of Different GETV Strains in Vero, C6/36 and U4.4 Cells

Replication kinetics of the GETV strains were further assessed in Vero, C6/36, and U4.4 cells following infections at MOI of 0.1. Extracellular RNA levels of various virus strains at 0, 8, 24, 48, 72, 96 hpi were determined, as shown in Figure 3. The replication rates and mean maximum titers of the GETV strains were summarized in Table 2. In Vero cells, the virus titers of GI MM2021, GIII MI-110, and GIII 14-I-605 increased exponentially within the first 8 hpi. In contrast, the GIV B254 showed delayed exponential increase in virus titer to after 8 hpi. While the GI MM2021 reached a plateau in titer at 24 hpi, other GETV strains attained a plateau at 48 hpi (Figure 3A). From 48 hpi onwards, the GIII MI-110 and GIII 14-I-605 showed significantly higher virus titers than those of GI MM2021 (1.6–2.3-fold) (Bonferroni posttest, *p* < 0.05 and *p* < 0.01, respectively) and GIV B254 (2.4–5.3-fold) (Bonferroni posttest, *p* < 0.001). On the other hand, GI MM2021 showed significantly higher virus titers than those of GIV B254 at 24 hpi (Bonferroni posttest, *p* < 0.05) and 48 hpi (Bonferroni posttest, *p* < 0.01) (Figure 3B). Regression analysis estimated that the GIII MI-110 and 14-I-605 replicated at a higher replication rate than those of GI MM2021 (1.9–2.4-fold) and GIV B254 (3.3–4.0-fold). The GIII MI-110 and 14-I-605 strains replicated at 3.0 × 10^5^ ± 7.2 × 10^4^ RNA copies/µL/day and 3.8 × 10^5^ ± 4.5 × 10^4^ RNA copies/µL/day, respectively, while the GI MM2021 and GIV B254 replicated at 1.6 × 10^5^ ± 3.5 × 10^4^ RNA copies/µL/day and 9.1 × 10^4^ ± 1.5 × 10^4^ RNA copies/µL/day, respectively (Table 2). All the GETV strains achieved their mean maximum titers at different time points post-infection in Vero cells (Table 2). The GIV B254 had the least mean maximum titer (7.72 × 10^6^ RNA copies/µL), followed by GI MM2021 (1.83 × 10^7^ RNA copies/µL), GIII 14-I-605 (3.12 × 10^7^ RNA copies/µL), and GIII MI-110 (3.27 × 10^7^ RNA copies/µL) (Table 2). 

In C6/36 cells, the virus titers of each GETV strain increased steadily over time (Figure 3A). The GIV B254 showed lower virus titer than those of GI MM2021, GIII MI-110, and GIII 14-I-605 throughout the infection (Figure 3B), with significant differences observed at 48 hpi onwards (Bonferroni posttest, *p* < 0.01). The GI MM2021, GIII MI-110 and 14-I-605 replicated at a similar rate at 3.5 × 10^4^ ± 3.6 × 10^3^, 3.2 × 10^4^ ± 3.9 × 10^3^, and 3.4 × 10^4^ ± 2.6 × 10^3^ RNA copies/µL/day, respectively, while the GIV B254 strain recorded a relatively 3.2–3.5-fold lower replication rate at 1.0 × 10^4^ ± 3.6 × 10^3^ RNA copies/µL/day (Table 2). In C6/36 cells, all GETV strains attained their mean maximum titers at 96 hpi. The GIV B254 strain showed the lowest mean maximum titer, 9.25 × 10^5^ RNA copies/µL, which was 2.8–3.3-fold lower than those of the other three GETV strains, which ranged from 2.63 × 10^6^ to 3.02 × 10^6^ RNA copies/µL (Table 2).

All four GETV strains did not replicate very well in U4.4 cells in comparison to the Vero and C6/36 cells. Virus titers of all virus strains increased exponentially within 8 hpi and declined steadily after peaking at 24 hpi (Figure 3A). Both GIII MI-110 and 14-I-605 showed significantly higher virus titers than those of GI MM2021 (2.5–6.3-fold) (Bonferroni posttest, *p* < 0.001) and GIV B254 (2.3–5.4-fold) (Bonferroni posttest, *p* < 0.001 and *p* < 0.01, respectively) at 24 hpi and onwards (Figure 3B). The virus replication rates in U4.4 cells were estimated based on the virus growth curve from 0 to 24 hpi due to the virus titer drop after 24 hpi. The GIII MI-110 and 14-I-605 recorded a relatively 2.7–5.8-fold higher replication rates at 1.8 × 10^4^ ± 763.4 and 1.1 × 10^4^ ± 3.2 × 10^3^ RNA copies/µL/day, respectively, in comparison to the GI MM2021 and GIV B254, which replicated at 3.1 × 10^3^ ± 1.2 × 10^3^ and 4.1 × 10^3^ ± 1.3 × 10^3^ RNA copies/µL/day, respectively (Table 2). All four GETV strains achieved the mean maximum titer at 24 hpi, ranging from 1.10 × 10^5^ RNA copies/µL to 4.35 × 10^5^ RNA copies/µL (Table 2).

In order to validate the infectivity of the extracellular viral samples, a plaque assay was performed to measure the infectious virus titers for selected time points during the exponential phase of infections (Appendix A). Overall, all GETVs showed increase in the infectious titers in all infected cells. Consistently, the GIV B254 showed lower infectious virus titers than those of GI MM2021, GIII MI-110, and GIII 14-I-605 at 48 hpi in the infected Vero and C6/36 cells, and at 24 hpi in the U4.4 cells (Appendix A).

### 3.4. Cytopathic Effects of GETV Infections in Vero, C6/36, and U4.4 Cells

Cytopathic effects (CPE) of all GETV strains in the Vero, C636, and U4.4 cells were observed at each time point. Figure 4 shows the morphology of the infected cells at 48 hpi. The GETV infections caused apparent CPE in the Vero cells, where most of the infected cells shrunk, rounded, and detached from the surface of the well (Figure 4). The degree of CPE caused by the GETV GI MM2021, GIII MI-110 and GIII 14-I-605 were much more pronounced in comparison to the GETV GIV B254, which induced relatively weaker CPE in Vero cells, in that relatively fewer cells had shrunk and detached from the surface of the wells. In C6/36 cells, all four GETV strains showed relatively moderate CPE in comparison to those induced in the Vero cells. The degree of CPE was similar for all virus strains, as evidenced by cell shrinkage and cell detachment. Nevertheless, it was noticed that the GETV GIV B254-infected cells showed less pronounced CPE in comparison to the others. On the other hand, all GETV infections in U4.4 did not cause any apparent CPE, the morphology of the infected cells looked similar to that of mock-infected cells at 48 hpi.

## 4. Discussion

Of the four major phylogenetic groups of GETV, GIII and GIV were the most recent circulating and geographically expanding virus groups. However, to date, the GETV GIII has been the sole lineage that was associated with manifestation of diseases in animals [5,10,13]. In this study, we examined and compared the genomic and in vitro phenotypic characteristics between the epidemic and non-epidemic GETV strains. While both epidemic GETV GIII strains consistently replicated at higher rates and produced higher virus titers in all cell lines, the non-epidemic GETV GIV strain showed the lowest replication rate and virus titer during infection. Our findings suggest that the phenotypic differences between the different GETV groups could be attributed to the genotypic variations unique to their respective groups, particularly those resulting in the non-conservative amino acid substitutions in the nsP3 and E2 proteins. 

The Japanese GETV MI-110 was among the first strains of GIII lineage that emerged and caused an outbreak of infection in horses in 1978, at Miho Training Centre, Ibaraki Prefecture, Eastern Japan [5]. In 2014, a recurrent outbreak caused by the GETV GIII 14-I-605 strain occurred among vaccinated racehorses at the same training center [13]. Sequence analyses between these two virus strains suggested the potential importance of the amino acid substitutions in the hypervariable domain (HVD) region of nsP3, which includes the T416P reported in this study, on the virological properties of the virus [5,24,25]. The nsP3 protein has two conserved domains and a HVD region; the latter is crucial for the interactions with host factors and plays an essential role in virus replication in the mosquito vectors and vertebrate hosts [26,27]. Thus, the genetic variations in this gene region may probably influence the virus replication competency in a particular host. In this study, both GIII GETV MI-110 and 14-I-605 strains exhibited higher replication rates and produced higher virus titers than the non-GIII strains in the mosquito and mammalian cells. This suggests that the GIII GETV undergoes an infection cycle more rapidly, thus infecting a greater number of cells and causing more CPE within the same period of infection, compared to the non-GIII strains. A virulence characteristic allowing the virus to replicate to a sufficient virus load before the onset of robust host immune response could be an important key advantage for the GIII GETV strains. This may also suggest the higher competency of the GIII strains in spreading from the initial infection site to other target tissues and organs where pathogenicity was observed in the infected hosts.

The GETV GIV B254 strain is a new virus strain recently isolated from *Culex fuscocephalus* in Malaysia, since the first virus isolation in 1955 [19]. It is phylogenetically distinct from the old Malaysian GETV MM2021, but similar to other GIV strains, where it shared the closest relationship with the China YN12031 strain isolated in 2012 [19,28]. It has been hypothesized that the GETV GIII and GIV viruses evolved from the GII Sagiyama strain. However, in comparison to the GIII viruses, the GIV viruses showed excessive amino acid substitutions not only in the nsP3 but also in the structural genes. This suggests that the GIV lineage may be under a different selection pressure potentially caused by differences in hosts. 

So far, both Malaysian GETV MM2021 and B254 have not been associated with any disease outbreaks in animals or humans. In our study, the GETV B254 demonstrated a relatively lower replication competence in all the cell lines used, as shown by the slower rate of replication and lower virus titers produced, compared to those of the GIII GETV strains. Relatively lesser CPE and dead cells were observed in the GETV B254-infected Vero and C6/36 cells through microscopic examination; however, further experiments are desired in future to quantitatively determine the degree of CPE caused by different GETV strains for better comparisons. Nevertheless, like the other strains with reduced virulence, GETV B254 formed plaques of much smaller sizes [29,30,31,32]. Evidently, these phenotypes suggest that the GIV B254 undergo a longer delay for virus replication and release, and consistent with a longer elapsed time between the successive infection cycles. As such, the GIV B254 strain is unable to effectively infect a large number of tissue cells and cause CPE that result in manifestation of disease. This also means that the GIV viruses could be transmitted between the mosquitoes and vertebrate hosts in nature without being detected due to the absence of disease. In relation, the GETV/SW/Thailand/2017 belonging to the GIV group was isolated from pig serum during a sero-surveillance in Thailand, where no disease was reported [20]. It is worth noting that the pig-origin GIV strain, in comparison to the mosquito-origin GIV viruses, showed an amino acid substitution at the E2 (L269V), which was exclusively associated with GIII lineage and was found to be the sole positive selection site in the structural genes (Table 1). As the E2 of alphaviruses has been associated with host range and pathogenicity [33], the substitution in this gene could possibly mark an adjustment towards acquisition of epidemic potential of the GIV virus strains, possibly resembling that of the A226V substitution in Chikungunya virus which resulted in a pandemic [34]. 

The first discovered Malaysian GETV MM2021 (1955) was of the GI lineage [4]. Between 1960s to 1970s, GETV was associated with large domestic animals in Malaysia, where the carabaos, horses and pigs showed the highest serological prevalence of infection [35,36]. The virus infections in these animals, however, were mostly inapparent. Isolation of several Malaysian GETVs from various mosquito species was reported during the same period. These viruses, of which the molecular characters were unknown, could be the other strains of the GI lineage which may be associated with mild or asymptomatic infections in the vertebrate hosts. In this study, the GETV MM2021 prototype strain showed replication efficiency comparable to the epidemic GETV GIII strains, although there were no common mutations between these viruses to explain this. Nevertheless, this could be caused by the in vitro adaptation of MM2021 strain to the cell culture after repeated and prolonged culture in the laboratory. This may lead to enhanced virus replication to produce higher virus titer, as previously seen in several other viruses [37,38,39]. 

The mammalian Vero cells and the *Aedes albopictus* C6/36 mosquito cells are the common susceptible cell lines used for arbovirus propagation and replication studies. The alphaviruses, such as CHIKV and SINV, have been shown to cause acute, lytic infection in the mammalian cells leading to strong CPE and apoptosis, while inducing persistent infection accompanied by lower virus titers in the mosquito cells [40,41]. These different infection dynamics were probably attributed to the spatial and temporal differences of virus replication and assembly process in the different types of cells [42,43]. Similarly, in our study, the GETV replicated to higher virus titers and caused more pronounced CPE in Vero cells than in the *Aedes albopictus*-origin C6/36 and U4.4 cells. While the C6/36 cells are lacking an intact RNA interference (RNAi) defense mechanism [44], the U4.4 cells are RNAi competent, thus, making it a better cell model for a more accurate presentation of the alphavirus infection in nature. Our findings showed an early declining titer of all GETV strains during infection in the U4.4 cells, indicating the virus growth restriction most likely by the RNAi response. Nevertheless, the epidemic GETV GIII strains have consistently exhibited higher replication competence even in this cell, in comparison to the non-epidemic GETV strains. Further investigations in mosquitoes are needed to better characterize the in vivo vector competence of the different GETV strains.

In summary, we compared the genetic and in vitro phenotypic characteristics between the epidemic and non-epidemic GETV. Several amino acid substitutions specific to the GETV GIII and GIV viruses in the nsP3 and E2 genes were identified. These amino acid substitutions may play a role in the higher replication rates, higher virus titers, and more pronounced CPE of the epidemic GIII viruses, compared to the non-epidemic viruses of GI and GIV groups. This further suggests that the higher virus replication competency to produce high virus titer during an infection may be the crucial determinants of virulence and epidemic potential of GETV. An in vivo study using a suitable animal model would be desired to further confirm the pathogenicity differences between the epidemic and non-epidemic GETV strains. 

## Figures and Tables

**Figure 1 viruses-14-00942-f001:**
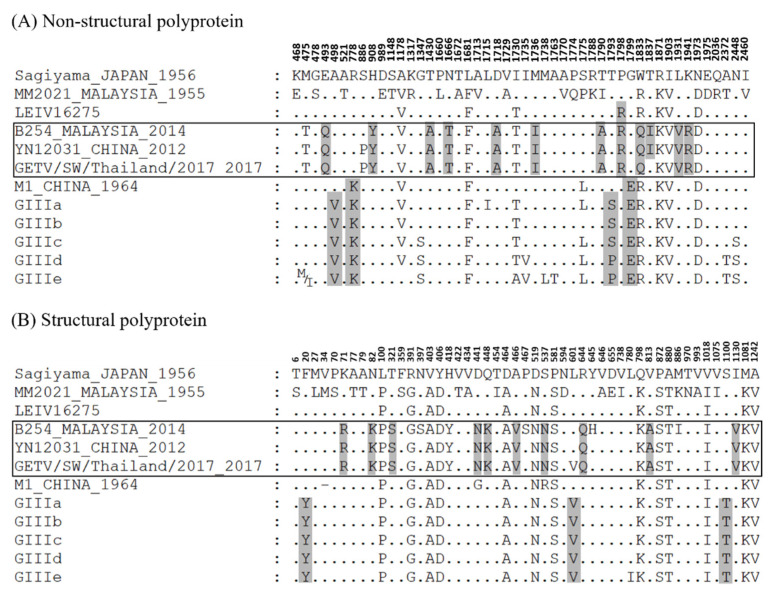
Comparison of (**A**) non-structural and (**B**) structural polyprotein amino acids between different groups of GETV. The dots represent similar amino acids in comparison to the GII GETV Sagiyama virus. The shaded areas represent the amino acid substitutions exclusive to the GETV GIII and GIV (in the box). The GETV GIII with homologous sequences are clustered together and indicated as GIIIa, GIIIb, GIIIc, GIIId, and GIIIe.

**Figure 2 viruses-14-00942-f002:**
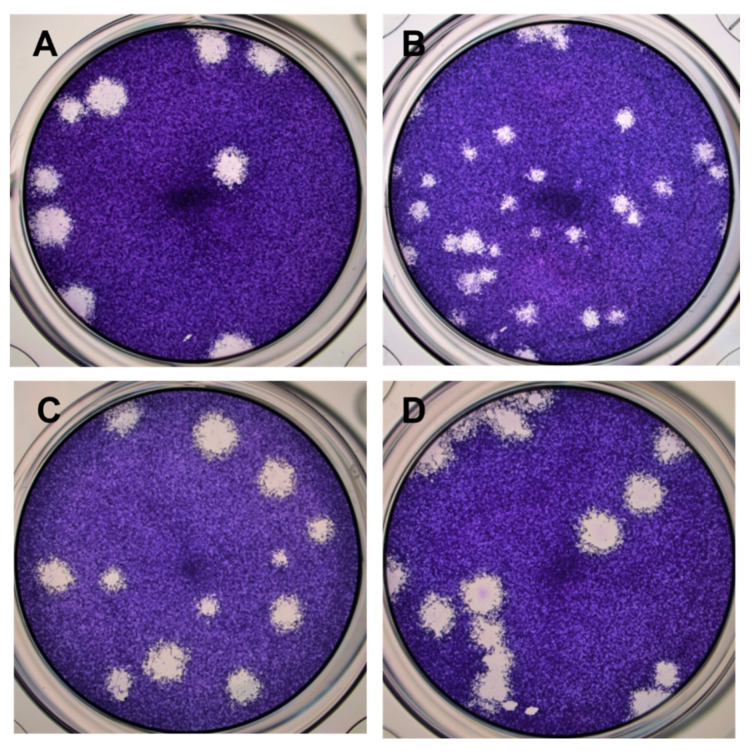
Plaque morphology of various GETV strains in Vero cells. Plaques shown were at 72 hpi; (**A**) GI GETV MM2021, (**B**) GIV GETV B254, (**C**) GIII GETV MI-110, (**D**) GIII GETV 14-I-605 (5× magnification).

**Figure 3 viruses-14-00942-f003:**
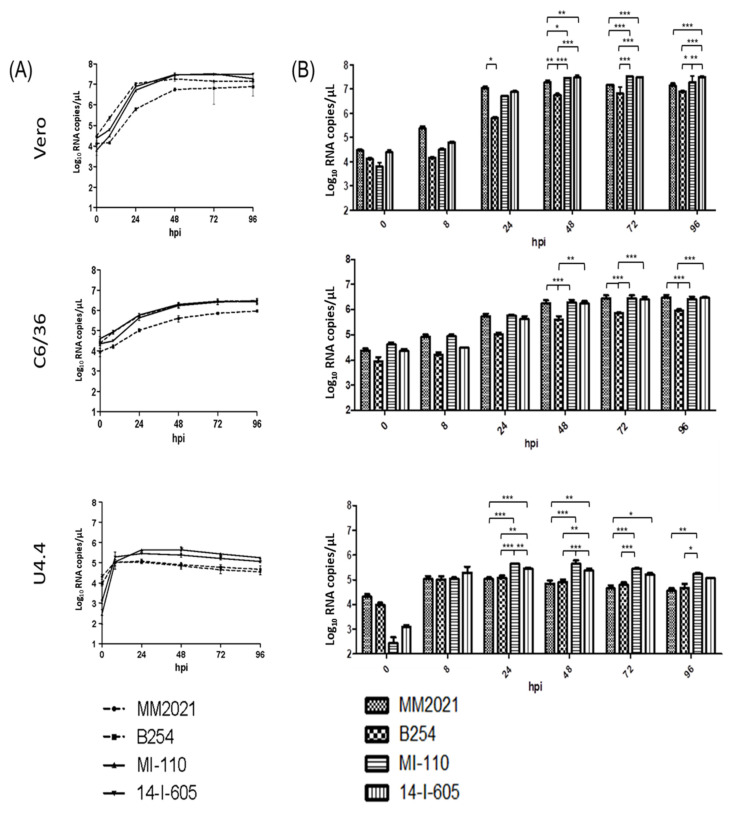
Replication kinetics of various GETV strains in Vero, C6/36 and U4.4 cells. Cells were infected with GETV at MOI = 0.1. Extracellular virus RNA levels at 0, 8, 24, 48, 72, and 96 hpi were quantitated using qRT-PCR. (**A**) Growth curves of different GETV strains in respective cell lines. (**B**) Bar graphs represent the growth of GETV in the cells. Data plots show the mean viral RNA copies and standard deviation (SD) of three independent replicates. Error bars indicate SD. Asterisks indicate statistical significance (* *p* < 0.05, ** *p* < 0.01, *** *p* < 0.001) as determined by Bonferroni test.

**Figure 4 viruses-14-00942-f004:**
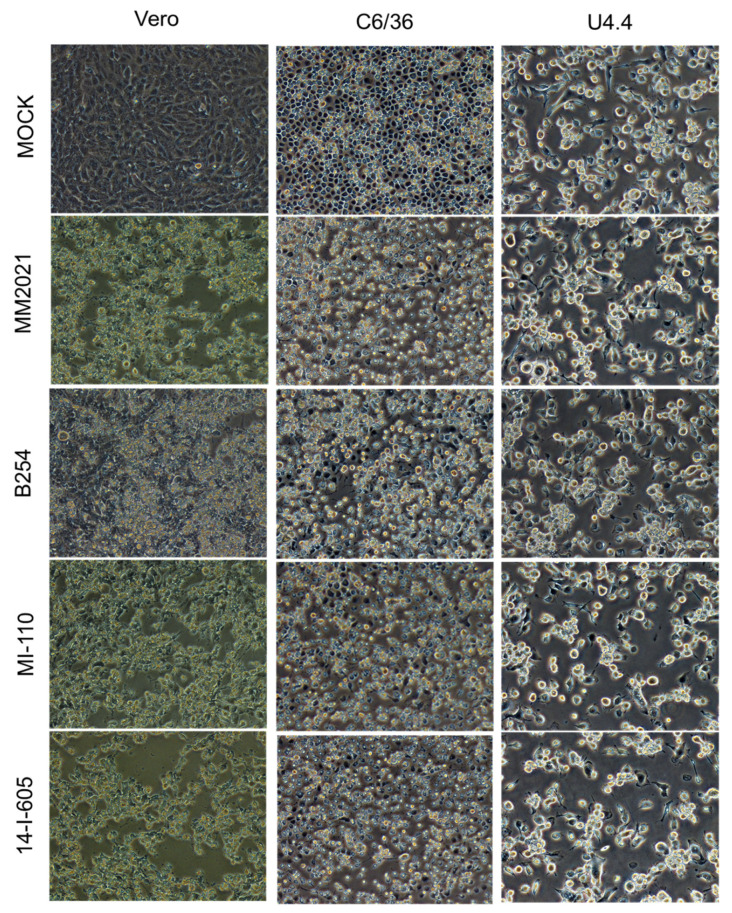
Cytopathic effects of GETV infection in Vero, C6/36, and U4.4 cells. Cells were infected with various strains of GETV at MOI of 0.1 and cytopathic effects (CPE) was observed at 48 hpi (200× magnification). Relatively more apparent CPEs were observed in the GETV-infected Vero and C6/36 cells, where the cells were shrunk, rounded, and refractile-appearing as compared to the larger and dark-appearing cells in the mock infections. No apparent CPE was observed in the GETV-infected U4.4 cells.

**Table 1 viruses-14-00942-t001:** Structural and non-structural protein amino acid substitutions specific to GETV GIII and GIV.

Protein	Amino Acid Position ^a^	Amino Acid Position ^b^	GIII	GI, II, IV	Conservative/ Non-Conservative
NSP1	498	498	V	A	Semi-Conservative
NSP2	778	244	K	R	Conservative
NSP3	1793	461	S/P	T	Non-Conservative (T→P)
NSP3	1799	467	E	G	Non-Conservative
C	20	20	Y	F	Conservative
E2	601	269	V	L	Conservative
E1	1100	285	T	S	Conservative
			GIV ^c^	GI, II, III	CONSERVATIVE/ NON-CONSERVATIVE
NSP1	493	493	Q	E	Conservative
NSP2	908	374	Y	H	Non-Conservative
NSP3	1430	98	A	T	Conservative
NSP3	1666	334	T	N	Conservative
NSP3	1718	386	A	D	Non-Conservative
NSP3	1736	404	I	M	Conservative
NSP3	1790	458	A	T	Conservative
NSP3	1798	466	R	P	Non-Conservative
NSP3	1833	501	Q	R/W	Non-Conservative (W→Q)
NSP3	1837	505	I	T	Non-Conservative
NSP4	1931	75	V	L	Conservative
NSP4	1941	85	R	K	Conservative
C	71	71	R	K	Conservative
C	82	82	K	N	Conservative
E3	321	53	S	T	Conservative
E2	441	109	N	D/G	Non-Conservative
E2	448	116	K	Q	Conservative
E2	466	134	V	A	Semi-Conservative
E2	537	205	N	S/R	Non-Conservative (S→N)
E2	644	312	Q	R	Semi-Conservative
6K	813	59	A	V	Semi-Conservative
E1	1130	315	V	I	Conservative

^a^ Position by non-structural polyprotein (nsp1-2-3-4) OR structural polyprotein (C-E3-E2-6K-E1), ^b^ position by gene, ^c^ all GETV GIV strains except LEIV 16275 Mag_RUSSIA_2000.

**Table 2 viruses-14-00942-t002:** Replication rates and mean maximum titers of various GETV strains in Vero, C6/36, and U4.4 cells.

Cell Line	GETV Strain	Replication Rate (RNA Copies/µL/Day)	Mean Maximum Titer (RNA Copies/µL)/hpi
Vero	MM2021	1.6 × 10^5^ ± 3.5 × 10^4^	1.83 × 10^7^/48 hpi
	B254	9.1 × 10^4^ ± 1.5 × 10^4^	7.72 × 10^6^/96 hpi
	MI-110	3.0 × 10^5^ ± 7.2 × 10^4^	3.27 × 10^7^/72 hpi
	14-I-605	3.8 × 10^5^ ± 4.5 × 10^4^	3.12 × 10^7^/96 hpi
C6/36	MM2021	3.5 × 10^4^ ± 3.6 × 10^3^	3.02 × 10^6^/96 hpi
	B254	1.0 × 10^4^ ± 554.4	9.25 × 10^5^/96 hpi
	MI-110	3.2 × 10^4^ ± 3.9 × 10^3^	2.63 × 10^6^/96 hpi
	14-I-605	3.4 × 10^4^ ± 2.6 × 10^3^	2.90 × 10^6^/96 hpi
U4.4	MM2021	3.1 × 10^3^ ± 1.2 × 10^3^	1.10 × 10^5^/24 hpi
	B254	4.1 × 10^3^ ± 1.3 × 10^3^	1.20 × 10^5^/24 hpi
	MI-110	1.8 × 10^4^ ± 763.4	4.35 × 10^5^/24 hpi
	14-I-605	1.1 × 10^4^ ± 3.2 × 10^3^	2.81 × 10^5^/24 hpi

## Data Availability

The data presented in this study are available in the article or Appendix A.

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
