# Peer review of "Genomic and In Vitro Phenotypic Comparisons of Epidemic and Non-Epidemic Getah Virus Strains"

_viruses, 2022, doi:10.3390/v14050942_

Round 1

Reviewer 1 Report

In this manuscript authored compared non-epidemic and epidemic Getah viruses based on their genomic sequence, replication rates and cytopathic properties. Authors identified several amino acid substitutions in viral proteins nsP3 and E2. They also demonstrated that epidemic strain have higher replication rates and more profound cytopathic properties compared to no-epidemic Getah. This is an insightful study, however it has several issues that need to be addressed prior to publication:

  1. Figure 2 shows plaques that seem to have different sizes, however it is unclear whether this difference is significant. Can authors measure the plaque sizes across at least three independent experiments and perform statistical comparison.
  2. In figure 3 authors used qRT-PCR for extracellular viral RNA to assess viral replication in mosquito and vertebrate cells. However, this approach is suboptimal. qRT-PCR will detect not only RNA from infectious viral particles, but also RNA in defective particles and viral RNA leaked from necrotic cells. Authors should instead perform plaque-assay to ensure that only infectious viral particles are detected.
  3. Figure 4 has a similar issue to figure 2 and is purely descriptive. Authors should quantify cell death caused by each virus using MTT or similar assay and determine the significance of the observed difference in CPE.

Reviewer 2 Report

The manuscript entitled “Genomic and in vitro phenotypic comparisons of epidemic and non-epidemic Getah virus strains” addresses the differences observed in the replication and cytopathic effects of Groups I, III, and IV Getah virus strains, including amino acid substitutions that may be related to potential virulence.

Overall, the manuscript is relatively well written, but grammar could be improved. Some suggested modifications are written directly on the manuscript. Minor errors, e.g., line 90, are also written directly on the manuscript.

Line 108: need to identify the “appropriate conditions” or indicate a reference that lists the conditions that were used.

Line 111: what kind of “maintenance media”. Identify and list the source of the media.

Line 131: indicate how the medium from each well was removed and if the wells were rinsed.

Line 188: spell out MOI first usage.

Figure 3: While the figure likely demonstrates the progression of replication and cytopathic effects, they are too small, including the legends, to read/interpret.

Figure 4: Based on the printout, it is very difficult to note differences in the cytopathic effects. The legend should indicate what the “arrows” point to. Again, very small and not very reader friendly.

References: numerous small errors (see comments on the attached manuscript). Carefully review and correct any errors that were not cited.

Round 2

Reviewer 1 Report

The authors have addressed all of my concerns and improved the quality of the manuscript. I have no further questions to the authors and believe that the manuscript can be published.